# Design and Control of a Single-Leg Exoskeleton with Gravity Compensation for Children with Unilateral Cerebral Palsy

**DOI:** 10.3390/s23136103

**Published:** 2023-07-02

**Authors:** Mohammadhadi Sarajchi, Konstantinos Sirlantzis

**Affiliations:** 1School of Engineering, University of Kent, Canterbury, CT2 7NT, UK; 2School of Engineering, Technology and Design, Canterbury Christ Church University, Canterbury, CT1 1QU, UK; konstantinos.sirlantzis@canterbury.ac.uk

**Keywords:** assistive robot, cerebral palsy, dynamic systems and control, gravity compensator, impedance control, kinematic analysis, lower-limb exoskeleton, unilateral exoskeleton, wearable robot

## Abstract

Children with cerebral palsy (CP) experience reduced quality of life due to limited mobility and independence. Recent studies have shown that lower-limb exoskeletons (LLEs) have significant potential to improve the walking ability of children with CP. However, the number of prototyped LLEs for children with CP is very limited, while no single-leg exoskeleton (SLE) has been developed specifically for children with CP. This study aims to fill this gap by designing the first size-adjustable SLE for children with CP aged 8 to 12, covering Gross Motor Function Classification System (GMFCS) levels I to IV. The exoskeleton incorporates three active joints at the hip, knee, and ankle, actuated by brushless DC motors and harmonic drive gears. Individuals with CP have higher metabolic consumption than their typically developed (TD) peers, with gravity being a significant contributing factor. To address this, the study designed a model-based gravity-compensator impedance controller for the SLE. A dynamic model of user and exoskeleton interaction based on the Euler–Lagrange formulation and following Denavit–Hartenberg rules was derived and validated in Simscape^™^ and Simulink^®^ with remarkable precision. Additionally, a novel systematic simplification method was developed to facilitate dynamic modelling. The simulation results demonstrate that the controlled SLE can improve the walking functionality of children with CP, enabling them to follow predefined target trajectories with high accuracy.

## 1. Introduction

CP is the most common child-onset mobility disorder [1,2], resulting from injury or damage to the brain before birth or in early childhood [3]. Individuals with CP suffer from inefficient and abnormal gait patterns [4], which can worsen with age, eventually losing ambulatory ability [5]. Abnormal gait patterns resulting from this condition increase the energy consumption in walking and lead to reduced mobility and independence [6,7]. Increased walking energy costs can mitigate engagement in society and negatively affect quality of life [8]. While there is no cure for CP, powered assistance from wearable LLEs has demonstrated significant potential in improving walking efficiency and reducing walking energy expenditure in impaired individuals [9,10,11]. One of the most common symptoms of CP is spasticity, which causes muscle stiffness and involuntary contractions [12]. LLEs can be used to assist children with CP in dealing with spasticity by providing external support to the affected muscles. The devices can help reduce spasticity by reducing the load on the affected muscles and providing assistance as needed [13]. Exoskeletons can also be programmed to provide resistance to certain movements to improve muscle strength and reduce spasticity over time [13].

Although numerous LLEs have been developed, only a limited number of LLEs have been tailored for the pediatric population, particularly for individuals with CP [10]. LLEs for children with CP can be classified into single-joint exoskeletons, multi-joint exoskeletons, and exoskeleton-plus-walker combinations. The ankle exoskeleton by Lerner et al. [14] and the knee exoskeleton by Lerner et al. [15] are single-joint exoskeletons with the most participants in their clinical studies, while WAKE-up as a multi-joint ankle–knee exoskeleton helped patients restore their physiological gait patterns, particularly at the ankle joint [16,17]. Meanwhile, Atlas2030 (multi-joint exoskeleton) [18] and Trexo (exoskeleton-plus-walker) [19] are commercialized devices in this field [11]. In exoskeleton-plus-walker devices, the walker carries the weight of the exoskeleton and ensures balance; however, the walker limits mobility and maneuverability, causing the patient to follow a fixed gait pattern, and their ability to maintain balance is not trained [20]. The literature reveals that brushless DC motors combined with harmonic drive gears are frequently employed as actuators in LLEs for children with CP. Furthermore, the literature indicates that finite state machines (FSMs), impedance controllers, and torque controllers are, respectively, the most common supervisory-level, high-level, and low-level controllers in the hierarchical control structure for LLEs designed for children with CP [11]. According to [11] (p. 2714), the average minimum age of CP users for different LLEs is 7.6 years old. Additionally, the data show that single-joint exoskeletons typically cater to GMFCS levels I–III, while multi-joint exoskeletons cover GMFCS levels I–IV, and exoskeleton-plus-walkers support GMFCS levels I–V.

SLEs, which only cover one side of the body, offer several advantages over LLEs, which cover both sides, for children with CP. SLEs can provide more targeted assistance to the affected leg or joint, which can improve walking patterns, balance, and overall mobility for children with asymmetrical gait patterns or weakness on one side of their body. Additionally, SLEs are typically lighter and less bulky than LLEs, which can increase comfort and ease of use for children with CP. This can lead to better compliance with wearing the device and greater flexibility in participating in a wider range of activities. SLEs are also more cost effective, versatile, and discreet than LLEs and can reduce stigma and promote greater social interaction for children with CP. This, in turn, expands the range of activities that these children can participate in. These features promise that SLEs have the potential to be a practical and effective option for children with CP who require assistive devices to improve their mobility and independence. Despite all of their advantages, to the best of the authors’ knowledge, no SLE has been specifically designed and prototyped for children with CP, particularly those with unilateral CP.

While no SLE is available for children with CP, the renowned ReWalk Robotics Ltd. has commercialized a one-sided wearable robotic ankle exoskeleton called “ReStore Exo-Suit”, which has been successful in providing unilateral assistance for post-stroke gait training and rehabilitation to deal with gait abnormalities, such as drop foot, resulting from muscle weakness after stroke [21,22]. It is worth noting that drop foot is not only a common gait abnormality observed in post-stroke patients but also prevalent in individuals with unilateral CP, particularly those with spastic hemiplegia [4,23].

LLE control for individuals with neurological disorders remains an open challenge in wearable robotic exoskeletons [24]. Control algorithms must adapt to real-world scenarios, like gait initiation and walking on diverse terrains. For widespread use, control strategies should be robust, user-friendly, and allow self-implementation [25]. Several studies have reviewed, compared, and analyzed a wide spectrum of control strategies for LLEs [24,26,27,28,29,30,31]. Yan et al. [26] categorized a variety of assistive control strategies for exoskeletons, of which predefined gait trajectory control and model-based control are the most popular. Predefined gait trajectory control is straightforward to action but forces the wearer to gait in a reference trajectory that may not be their normal one. Model-based control is widely used in many applications, but it requires a precise dynamic model of the coupled human–exoskeleton system [26]. When it is challenging to derive an accurate dynamic model of a human–exoskeleton system, fuzzy control can be applied to present an intuitive understanding of how to deal with human–exoskeleton interaction [26]. Moreover, predefined gait trajectory controllers are usually employed by LLEs for individuals with full paraplegia or severe mobility disorders [32]. Model-based controllers are widely used by LLEs, assisting users with mild or moderate paraplegia or muscle weakness who actively participate in the walking process [32]. Although there are several control strategies for assistive robotics, model-based control is a common practice in humanoid robotics [33,34].

Relying on human–exoskeleton dynamic models, model-based control strategies can provide direct access to the gravity vector of the model, which can then be effectively utilized for gravity compensation [35]. In the gravity compensation scenario, an LLE assists its user in following a standard gait trajectory, while the user never feels the extra weight and inertia of the LLE. While a gravity compensator may be less crucial for lightweight single-joint exoskeletons, it is critical for multi-joint LLEs, which have significantly more weight. If gravity is not compensated for high-mass LLEs, the level of assistance provided to the user is reduced [32]. Some studies have benefited from gravity-compensator advantages in their LLE controller design [32,36,37]. However, only a very limited number of studies have employed this strategy to fully compensate for the gravity impact on multi-joint LLEs, which need complex dynamic modelling [38].

Children with CP consume more metabolic energy during walking in comparison to their healthy peers [39,40,41]. The exoskeleton weight and mass distribution directly affect metabolic consumption and functional performance. Studies have shown that carrying additional load mass can significantly increase metabolic energy consumption during overground walking [42,43,44]; thus, the use of gravity compensation in model-based control strategies is highly beneficial for LLEs designed for children with CP. However, to the best of the authors’ knowledge, no study has designed and evaluated the effect of a gravity-compensator controller for LLEs in the pediatric population with CP, which is implemented in this paper.

This research makes the following significant contributions: (a) it introduces and discusses in detail the design of a uniquely tailored hip–knee–ankle–foot unilateral exoskeleton, specifically developed for children aged 8 to 12 suffering from unilateral CP with GMFCS levels I–IV; (b) it presents extensive anthropometric analysis for both healthy and CP-affected children across the gender spectrum, considering different GMFCS levels and percentile rates, thus ensuring the device aptly caters to the varied body dimensions of its intended users; (c) the study presents analytically kinematics and dynamics modelling for this assistive exoskeleton, taking into account necessary gravity compensation; (d) the study introduces a novel method for simplifying the analytical exoskeleton dynamics model in the Simscape^™^ environment, thereby facilitating easier application of dynamic modelling. Additionally, a comprehensive evaluation of the derived dynamic model within both Simscape^™^ and Simulink^®^ environments confirms its effectiveness and demonstrates the controller’s efficacy. This thorough evaluation process represents a substantial advancement in the design of efficient mobility aids; (e) it proposes, after meticulous evaluation of multiple criteria, a specific type of motor and gears combination (brushless DC motors and harmonic drive gears) that yields improved overall system efficiency, with specific ratios assigned to each joint; and (f) in response to varying muscle tone and spasticity of children with CP, the proposed novel assistive device is equipped with distinct mechanical, sensory, and control elements (i.e., a pioneering telescopic structure and clamping mechanism, as well as force sensors integrated with the footplates), facilitating effortless size and torque adjustments to accommodate children with diverse body proportions and strength.

This paper is organized as follows. First, the mechanical and electronic design of the exoskeleton is presented. This is followed by kinematic analysis of the exoskeleton, which includes both forward and inverse kinematic models. In the fourth section, the dynamic modeling of the coupled human–SLE system is presented, and subsequently, the controller with a gravity compensator is designed in the next section. Simulation analysis of the controlled coupled human–SLE system follows. The final section concludes the paper with highlighted remarks and key findings.

## 2. The Pediatric Single-Leg Exoskeleton

This study aimed to design and control an SLE that can assist with the movement of the hip, knee, and ankle joints of one leg in children with unilateral CP. To achieve this goal, a lightweight and size-adjustable unilateral exoskeleton was developed. This prototype can be quickly adjusted to accommodate a wide range of heights, from 1 m to 1.6 m, making it suitable for children with unilateral CP aged 8 to 12 years old across GMFCS levels I–IV. The initial design of the SLE consisted of two actuated joints at the hip and knee, and a prototype was developed, as shown in Figure 1. To enhance the capabilities of the exoskeleton further, a second design was developed, which incorporated three actuated joints at the hip, knee, and ankle. This design was simulated, as shown in Figure 2. Although both designs have similarities, this study focused on the second design for further consideration due to its advanced capabilities. In Figure 1, the SLE is worn by a healthy adult with child-size body dimensions, standing at 157 cm and weighing 52 kg.

Table 1 provides a comprehensive overview of the average body dimensions for children with CP, including both girls and boys aged 8 and 12, accounting for the 50th percentile across GMFCS levels I–IV [45,46]. The table also illustrates the average body dimensions for TD children, encompassing both girls and boys aged 6 and 12 and accounting for the 50th percentile. Additionally, the table presents the minimum and maximum dimensions for the SLE to offer a better understanding of how the exoskeleton accommodates the leg and foot length requirements for a broad range of children with CP, aged 8 to 12 years old and across GMFCS levels I–V. Since detailed information about leg length (LL) and other measurements is not provided in [45,47,48], the height of children with CP is compared to that of their TD peers of the same height in [49] in order to estimate the leg and foot length of these children. To define the minimum size of the exoskeleton, the average body dimensions of 8-year-old girls with CP at GMFCS level IV are selected. For the maximum size of the exoskeleton, the average body dimensions of 12-year-old boys with CP at GMFCS level I are chosen. This table demonstrates that the exoskeleton leg length can be adjusted from 60 cm to 86 cm, fully accommodating a minimum user leg length of 62.8 cm and a maximum user leg length of 85.7 cm. Additionally, the table illustrates that the exoskeleton footplate length is 23.5 cm, effectively accommodating both the minimum and maximum user foot lengths, which are 16.5 cm and 21.7 cm, respectively.

Figure 2 shows the exoskeleton configuration, including actuators in the sagittal plane mounted on the joints, which are joined by adjustable links to the upper and lower leg. This figure also shows the exploded view for the actuators, drivers, links, and foot assembly.

### 2.1. Mechanical Design

The SLE is made of lightweight and durable materials, like aluminum tubes, to reduce the overall weight of the device (5.2 kg). The exoskeleton is equipped with telescopic cylinder mechanisms to ensure adjustable sizing and prevent misalignment and kinematic incompatibility between the user–exoskeleton joints, as children with CP come in various sizes, and their bodies can change over time (Figure 2D). The vertical cut was made on one end of the outer tube and a clamp was attached to its outside, enabling easy adjustment to the desired length (Figure 2D). Furthermore, the adjustment process is expedited and simplified by incorporating quick-release skewers (Figure 2D). These mechanisms on both the upper and lower leg provide maximum adaptability for a wide range of users and ensure that the appropriate level of support is provided.

The device’s braces come with adjustable straps and buckles, allowing for a customized fit and level of tightness to provide optimal support tailored to the unique needs of each user, as children with CP can have different and varying levels of muscle tone and spasticity. The SLE has removable and adjustable inserts and pads that can be added or removed to modify the level of support offered by the assistive device. For instance, a child may need extra support at their ankle joint, and an additional insert can be inserted to provide the necessary support. If needed, the exoskeleton design can include heel wedges to correct abnormal gait patterns. The use of padding and straps helps distribute pressure evenly and allows for easy adjustments, resulting in a user-friendly interface that provides comfort for both the user and caregiver while using the exoskeleton. The exoskeleton is equipped with an emergency stop button, a non-slip sole (Figure 2E), and locking mechanisms that prevent the knee joint from bending too far, reducing the risk of falls and protecting the child from potential injuries. These features improve the overall safety of the exoskeleton, ensuring a safe and comfortable experience for the user.

In the mechanical design of LLEs for children with CP, the range of motion is a critical parameter to consider. Children affected by CP usually exhibit abnormal muscle tone, potentially leading to a reduced range of motion, contractures, and joint deformities. Thus, the design of an exoskeleton needs to take into account the individual needs of each child to ensure the optimal range of motion. The designed SLE in this study provides a range of motion encompassing 30° of hip flexion and 90° of extension, 120° of knee flexion and full extension, as well as 45° of plantar flexion and 30° of dorsiflexion in the ankle joints. This enables the child to maintain a natural gait. The exoskeleton’s range of motion is adjustable to cater to individual needs and the stage of the child’s development as their range of motion is likely to change over time [13,50].

### 2.2. Actuation Unit

Actuators are essential components in the design of exoskeletons for children with CP, as they provide the necessary assistance to the child’s leg(s). Brushless DC (BLDC) motors are the most frequently cited type of actuator in the literature for powered LLEs designed for children with CP, owing to their lightweight, compact, accurate, reliable, silent, and sufficiently powerful attributes [11]. TD children can generate a maximum torque of 0.7 Nm/kg at the hip, 0.5 Nm/kg at the knee, and 1.2 Nm/kg at the ankle during overground walking [51,52]. Moreover, they can produce a maximum power of 0.8 W/kg, 0.7 W/kg, and 3 W/kg, respectively, at the hip, knee, and ankle while walking overground [51,52]. For a TD child weighing 30 kg, the hip joint requires a maximum torque of around 21 Nm and power output of 24 W, the knee joint needs approximately 15 Nm and 21 W, and the ankle joint requires about 36 Nm and 90 W during movement. However, a recent study that evaluated the mean and maximum joint torque, power, and velocity of TD children indicates that children can achieve significantly higher maximum values for joint torque and power compared to the values mentioned earlier [20]. The study’s findings suggest that these higher maximum values could be taken into consideration while selecting appropriate actuators for pediatric LLEs to provide optimal assistance.

Following an extensive review of the literature and analysis of various motors and reduction ratios to satisfy the specific criteria and essential requirements, an actuator unit was selected. The joint actuation unit is comprised of a high-performance 24 V BLDC motor (EC-flat, Maxon Motor AG, Sachseln, Switzerland) and a state-of-the-art strain-wave gearing system (CPU-17A-M, Harmonic Drive^®^ AG, Limburg, Germany). The EC-flat series of BLDC motors was chosen for its superior durability, lightweight construction, quiet operation, and high power efficiency, all of which render it an excellent option for use in powered exoskeletons. The particular gearing system, known as CPU-17A-M, is specifically designed to operate seamlessly with the BLDC motor, ensuring smooth and precise movement of the exoskeleton’s joint. The CPU-17A-M series is renowned for its compact size, high precision, and low backlash characteristics, making it an ideal choice for joint actuation units in pediatric exoskeletons.

The actuation unit configuration for the three actuated joints is illustrated in Table 2. The table demonstrates that each of the three actuation units is capable of covering the maximum power, torque, and velocity addressed in the literature. For instance, knee and ankle joints have similar actuators with a power rating of 150 W. However, the knee joint has a reduction ratio of 50, while the ankle joint has a reduction ratio of 120. Moreover, it has been observed that TD children can generate a maximum torque of 0.5 Nm/kg at the knee and 1.2 Nm/kg at the ankle during overground walking. The actuation units used in the exoskeleton’s knee and ankle joints can produce torques that are fully compatible with the structure of the knee actuator. Additionally, Figure 2A illustrates an exploded view of the strain-wave gearbox, with “WG” representing the wave generator. In order to drive each motor, Maxon EPOS4 50/15 controllers were selected (Figure 2C).

These controllers are specifically designed to support the EtherCAT protocol and operate as a slave in the network. Additionally, they offer high precision and reliability in controlling the motor’s speed, position, and torque. The selection of these controllers ensures optimal performance of the joint actuation unit, providing smooth and accurate motion control for the powered SLE. The combination of the BLDC motor, strain-wave gearing system, and high-quality EPOS4 driver results in a sophisticated and efficient actuation unit for the SLE, capable of meeting the demanding requirements of individuals with mobility impairments.

### 2.3. Sensory System

Several different sensors are utilized throughout the SLE to ensure optimal performance and control. These sensors include hall sensors, absolute encoders, pressure sensors, and inertial measurement units (IMUs).

Hall sensors are used to measure the position and speed of the motors, providing real-time feedback to the controller for optimal control of joint movement. This real-time feedback can help reduce spasticity by delivering a consistent and controlled movement pattern, reducing sudden or unexpected movements that can trigger spasms. In addition, the controller can adjust the level of assistance provided by the exoskeleton in real time based on the feedback from the hall sensors, further reducing the likelihood of spasticity.

To provide more accurate joint angle measurement, a magnetic absolute rotary encoder is integrated for each actuated joint (Figure 2A,B). The RLS AksIM-2™ magnetic encoder is directly connected to the Maxon EPOS 4 motor driver using the SSI communication protocol, with a high resolution of up to 20 bits. Absolute encoders offer extremely precise joint angle measurements, resulting in more natural and fluid movements for the exoskeleton while reducing the likelihood of spasms. By providing real-time feedback from hall sensors and highly accurate joint angle measurement from absolute encoders, the exoskeleton can facilitate natural and efficient walking patterns tailored to the user’s specific needs.

The foot of the exoskeleton is equipped with an electronic sensor board (Figure 2E) that tracks the orientation of the foot and monitors the ground reaction force. To accurately measure pressure distribution from multiple directions, the exoskeleton utilizes four FlexiForce A201 force sensors that are connected to the electronic board (Figure 2E). By monitoring pressure distribution from different directions, the exoskeleton can adjust its movement patterns to provide more natural and controlled movements, which can aid in preventing falls and improving overall safety during walking. The output signals of force-sensitive resistor (FSRs) are typically small, so four op-amp microchips (LF356) are recruited to amplify the force sensors’ signals and to make them suitable for an STM32 Nano microcontroller to process the signals and transmit them through high-speed CAN bus (Figure 2E).

The acquisition of foot orientation is carried out by an embedded IMU (MPU9250) located on the foot electronic board. By utilizing a combination of gyroscope, accelerometer, and magnetometer data, the IMU is able to measure the foot’s position and orientation with high precision. This information can be utilized to evaluate gait patterns and detect any asymmetries or abnormal movements, enabling healthcare professionals to customize treatment plans and interventions according to the unique requirements of each child. To ensure accurate gait phase detection, it is necessary to equip both legs with footplates that include electronic sensory boards. Furthermore, each footplate’s height is 23 mm, and since both legs are equipped with footplates with the same height, a user will not feel any difference in height between their legs. An EasyCAT EtherCAT Shield is recruited to receive sensory information from both feet through the CAN bus and actuator signals from a series of EPOS 4 drivers via EtherCAT cable to facilitate their transmission to the central controller (EtherCAT Master).

### 2.4. Control Architecture

To ensure precise control of all three motors and enable real-time sensor data collection, we established the control system configuration for the SLE, as illustrated in Figure 3. The control system in the SLE is typically divided into three essential components: local controllers (EtherCAT Slaves); sensory and communication system; and the central controller (EtherCAT Master). The first part involves local controllers providing low-level control for actuators and are located on the Maxon Epos 4 drivers, which are connected serially under EtherCAT communication protocol, beginning from the ankle driver and ending at the hip driver. The second component of the control system architecture consists of a sensory and communication system designed to receive data from local controllers and FSRs installed in the feet components of the SLE and transmit control signals from the central controller to local controllers. This system communicates with local controllers via EtherCAT and FSRs through CAN Bus. Finally, we selected a state-of-the-art real-time embedded controller (Unit real-time target machine, Speedgoat, Köniz, Switzerland) as the central control unit to ensure system stability and optimize performance.

The Unit real-time target machine is renowned for its compact size and ability to effectively execute high-performance control applications and conduct small-scale plant simulations. This central control unit is capable of serving as the EtherCAT master and executing the system with a frequency of 1 kHz. One of the key benefits of using Speedgoat is the ability to design and develop control algorithms in MATLAB Simulink^®^ and then compile them into a C executable file for deployment on the target machine. Due to the close and direct interaction between Speedgoat and MATLAB, all the SLE simulations and control designs were carried out in MATLAB R2021a, which is highly compatible with the installed software version on the central controller. To create an intuitive and user-friendly interface, a high-performance host computer was connected to the Speedgoat target machine via an Ethernet cable to serve as the human–machine interface (HMI).

The Unit real-time target machine enables the implementation of adaptive control algorithms that account for varying levels of spasticity and motor function impairments as defined by the GMFCS (Gross Motor Function Classification System). This adaptability ensures that the exoskeleton provides personalized assistance and support to each child based on their specific needs. Furthermore, Speedgoat real-time target machines also support hardware-in-the-loop (HIL) simulation, enabling the testing of control strategies in real time without risking damage to the hardware or causing discomfort to users. This feature is crucial when working with children with CP, ensuring their safety and comfort during the development and testing phases. Lastly, the system’s ability to handle multiple control loops and sensor inputs at different rates enables seamless integration and synchronization of various sensors and actuators, which is critical for addressing spasticity and coordinating the exoskeleton’s movement with the user’s intentions. By utilizing a Speedgoat real-time target machine to control SLEs for children with CP, researchers and engineers can develop more effective, personalized, and responsive assistive devices that improve the mobility and quality of life of affected individuals. Moreover, it is essential to emphasize that the SLE relies on tethered power assistance for its functionality. This method enables the device to maintain a stable and uninterrupted energy supply through a physical connection to a power source.

## 3. Kinematic Analysis

Kinematic analysis is essential for understanding the movement of SLEs and their interaction with the human body. It involves examining the relationship between the exoskeleton’s component position and orientation and the corresponding joint angles necessary for motion [53]. The presence of misalignment and kinematic incompatibility between the user–exoskeleton joints can lead to the generation of undesired human–robot interaction torques. This underscores the importance of performing a thorough kinematic analysis of SLEs, particularly for children with CP [54,55,56]. Kinematic analysis can help optimize the design of SLEs for children with CP by ensuring that the device provides the appropriate support and assistance based on the child’s individual movement patterns and needs. This can lead to improved mobility, function, and quality of life for these children [57].

Kinematic analysis is typically divided into two categories [35]: forward kinematics, which deals with determining the exoskeleton’s end-effector (such as the foot) position and orientation based on joint angles, and inverse kinematics, which involves calculating the joint angles needed to achieve a desired end-effector position and orientation. Forward and inverse kinematics are crucial for developing practical and comfortable SLEs that can restore mobility to people with lower-limb impairments.

### 3.1. Forward Kinematic

Forward kinematics involves using the joint angles and segment lengths to calculate the position and orientation of the end effector (e.g., the foot) relative to the starting position of the limb. This information can be used to predict how the limb will move under certain conditions, such as when walking or standing. By using forward kinematics, the movement of the exoskeleton joints can be translated into the movement of the foot, allowing for precise control and monitoring of the gait pattern [35,58]. Moreover, forward kinematics can help determine how the device should be designed to provide a user with appropriate support and assistance. In the case of SLEs for children with CP, forward kinematics can be used to tailor the exoskeleton to the individual child’s gait pattern and specific needs. By analyzing the child’s gait using motion capture or other techniques, the joint angles of the exoskeleton can be adjusted to provide a more natural and efficient gait pattern [54,59,60].

The Denavit–Hartenberg (D-H) matrix method is commonly used to derive forward kinematics [35], which provides insights into the range of motion, joint limits, and performance of the exoskeleton. Despite its lengthy and intricate mathematical process, the D-H method establishes a systematic framework for modelling the kinematics of robotic mechanisms [61]. In Figure 4, the kinematic configuration of an SLE is shown following the D-H standard form, with each link having specific frames assigned to it. In the figure presented, the starting point *O*_0_ is defined as a stationary base frame, while the terminal coordinate system *O*_3_ is denoted as the foot end effector. The kinematic transformation depicted in this figure adheres to the guidelines and principles outlined by the D-H model. This study entails the independent modelling of the exoskeleton in two distinct environments, namely Simscape and Simulink. As the two models must be compatible with each other, in Figure 4, the reference frame (*X*_0_, *Y*_0_, *Z*_0_) is chosen to be identical to the global frame (*X*, *Y*, *Z*), thereby ensuring consistent results using two diverse and autonomous methodologies.

According to Figure 4, Table 3 presents the tabulated form of the D-H parameters, where $\theta_{i}$ represents the joint angle, $d_{i}$ corresponds to the joint offset, $a_{i}$ denotes the link length, and $\alpha_{i}$ specifies the link twist of the *i*-th link. The transformation matrix of each link is provided below with reference to Table 3.
(1)$$T_{0}^{1}=\left[\begin{array}{cccc} cos(\theta_{1}-\frac{\pi }{2}) & -sin(\theta_{1}-\frac{\pi }{2}) & 0 & a_{1}\;cos(\theta_{1}-\frac{\pi }{2})\\ sin(\theta_{1}-\frac{\pi }{2}) & cos(\theta_{1}-\frac{\pi }{2}) & 0 & a_{1}\;sin(\theta_{1}-\frac{\pi }{2})\\ 0 & 0 & 1 & 0\\ 0 & 0 & 0 & 1 \end{array}\right]$$
(2)$$T_{1}^{2}=\left[\begin{array}{cccc} cos(\theta_{2}) & -sin(\theta_{2}) & 0 & a_{2}\;cos(\theta_{2})\\ sin(\theta_{2}) & cos(\theta_{2}) & 0 & a_{2}\;sin(\theta_{2})\\ 0 & 0 & 1 & 0\\ 0 & 0 & 0 & 1 \end{array}\right]$$
(3)$$T_{2}^{3}=\left[\begin{array}{cccc} cos(\theta_{3}+\frac{\pi }{2}) & -sin(\theta_{3}+\frac{\pi }{2}) & 0 & a_{3}\;cos(\theta_{3}+\frac{\pi }{2})\\ sin(\theta_{3}+\frac{\pi }{2}) & cos(\theta_{3}+\frac{\pi }{2}) & 0 & a_{3}\;sin(\theta_{3}+\frac{\pi }{2})\\ 0 & 0 & 1 & 0\\ 0 & 0 & 0 & 1 \end{array}\right]$$

Based on the derived equations, the forward kinematics model for the SLE can be expressed as follows:(4)$$\begin{array}{c} T_{0}^{3}=T_{0}^{1}T_{1}^{2}T_{2}^{3}\\ =\left[\begin{array}{cccc} cos(\theta_{1}+\theta_{2}+\theta_{3}) & -sin(\theta_{1}+\theta_{2}+\theta_{3}) & 0 & P_{x}\\ sin(\theta_{1}+\theta_{2}+\theta_{3}) & cos(\theta_{1}+\theta_{2}+\theta_{3}) & 0 & P_{y}\\ 0 & 0 & 1 & 0\\ 0 & 0 & 0 & 1 \end{array}\right] \end{array}$$
where $P_{x}$ and $P_{y}$ represent the position of frame 3, which is the forefoot of the exoskeleton (end effector), relative to the global frame, along the *x* and *y* axes, respectively. These position values are computed as follows:(5)$$P_{x}=a_{2}\;sin\left(\theta_{1}+\theta_{2}\right)+a_{1}\;sin\left(\theta_{1}\right)+a_{3}\;cos(\theta_{1}+\theta_{2}+\theta_{3})$$
(6)$$P_{y}=-a_{2}\;cos\left(\theta_{1}+\theta_{2}\right)-a_{1}\;cos\left(\theta_{1}\right)+a_{3}\;sin(\theta_{1}+\theta_{2}+\theta_{3})$$

The transformation matrices are utilized directly and effectively to derive the dynamic equations of the human–SLE system.

### 3.2. Inverse Kinematic

Inverse kinematics involves using the desired position and orientation of the end effector to calculate the joint angles required to achieve that position [35]. This can be particularly useful in designing exoskeletons for children with CP, as the exoskeleton can be programmed to assist with specific movement patterns that are difficult for the child to perform on their own. Moreover, the inverse kinematics model can help optimize the exoskeleton’s design, including the size and shape of links and joints, to maximize performance while minimizing energy expenditure and discomfort, which are critical for children with CP. This process involves simulating different gait patterns and evaluating the exoskeleton’s ability to reproduce them accurately [15,54,62].

When dealing with an SLE, a geometric approach towards inverse kinematics is considered highly suitable for determining the variables $\theta_{1}$, $\theta_{2}$, and $\theta_{3}$ that correspond to the foot’s position and orientation, which serves as the end effector. The determination of the inverse trigonometric function does not result in a unique solution, and consequently, it gives rise to multiple solutions in the context of robot inverse kinematics. To address this issue, a common approach involves incorporating the structural properties of the robot and the actual pose. It is established in the literature that an analytical solution for the inverse kinematics problem can be derived when the kinematic chain consists of five or fewer DOF. In such cases, the angle of each joint can be determined through an algebraic method [35,63,64].

The inverse kinematic problem of the SLE involves determining the joint angles needed to achieve the desired position (*X*, *Y*) and orientation (*Φ*) of the end effector (foot) (Figure 5). Specifically, (7)–(14) show the relationship between the joint angles and the desired position and orientation of the end effector. Equation (7) releases the *x* and *y* coordinates of the second link (shank) with respect to the desired position (*X*, *Y*) and orientation (*Φ*) of the end effector (foot).
(7)$$\left\{\begin{array}{c} x=X-a_{3}\sin\Phi \\ y=Y-a_{3}\cos\Phi \end{array}\right.$$

Equation (8) gives the *x* and *y* coordinates of the second link (shank) in terms of the joint angles *θ*₁ and *θ*₂, where *a*₁ and *a*₂ are the lengths of the first (thigh) and second (shank) links, respectively.
(8)$$\left\{\begin{array}{c} x=a_{1}\sin\left(\theta_{1}\right)+a_{2}\sin\left(\theta_{1}+\theta_{2}\right)\\ y=a_{1}\cos\left(\theta_{1}\right)+a_{2}\cos\left(\theta_{1}+\theta_{2}\right) \end{array}\right.$$

By squaring ***x*** and ***y*** and adding them, we obtain (9):(9)$$x^{2}+y^{2}=a_{1}^{2}+a_{2}^{2}+2a_{1}a_{2}\cos\left(\theta_{2}\right)$$

By solving (9) for *θ*₂ using the inverse cosine function, the joint angle *θ*₂ can be determined.
(10)$$\theta_{2}=\pm {\;\cos}^{\;1}\left(\frac{x^{2}+y^{2}-a_{1}^{2}-a_{2}^{2}}{2a_{1}a_{2}}\right)$$

In the case of the SLE, only the negative value of *θ*₂ is acceptable. Equation (11) gives the expression for *θ*₂ in terms of the desired position (*X*, *Y*) and orientation (*Φ*) of the end effector, where *a*₃ is the length of the third link (foot).
(11)$$\theta_{2}=-\cos^{-1}\left(\frac{{\left(X-a_{3}\sin\left(\Phi \right)\right)}^{2}+{\left(Y-a_{3}\cos\left(\Phi \right)\right)}^{2}-a_{1}^{2}-a_{2}^{2}}{2a_{1}a_{2}}\right)$$

Moreover, for $\theta_{1}$, we obtain the following:(12)$$\theta_{1}=\gamma -\alpha =\tan^{-1}\left(\frac{x}{y}\right)-\tan^{-1}\left(\frac{a_{2}\sin\left(\theta_{2}\right)}{a_{1}+a_{2}\cos\left(\theta_{2}\right)}\right)$$

Equation (13) gives the expression for *θ*₁ in terms of the joint angles *θ*₂ and the desired position (*X*, *Y*) and orientation (*Φ*) of the foot.
(13)$$\theta_{1}=\tan^{-1}\left(\frac{X-a_{3}\sin\left(\Phi \right)}{Y-a_{3}\cos\left(\Phi \right)}\right)-\tan^{-1}\left(\frac{a_{2}\sin\left(\theta_{2}\right)}{a_{1}+a_{2}\cos\left(\theta_{2}\right)}\right)$$

Finally, (14) gives the expression for the joint angle *θ*₃ as follows:(14)$$\theta_{3}=\Phi -\theta_{1}-\theta_{2}$$

It is crucial to emphasize that the desired path should be determined by a rehabilitation plan created by a healthcare professional treating the child with CP. Consequently, corresponding inverse kinematic analysis was conducted to improve the understanding of the SLE’s motion capabilities.

## 4. Dynamic Modelling

In this section, the dynamic model of the coupled human–SLE system is derived using the transformation matrices discussed in the previous section. This helps enhance the understanding of the complex behavior of the coupled human–SLE system and enables the design of a more effective control algorithm. As a result, smoother and more coordinated movements are achieved, preventing the triggering of spasms in CP users. This dynamic model effectively aids in designing the gravity compensator within the control system, considering the complex interaction between the user and the exoskeleton. The gravity compensator minimizes or eliminates the exoskeleton’s weight impact on the user, reducing energy expenditure and making the device feel lighter and more comfortable to wear. As mentioned earlier, children with CP have twice the metabolic consumption compared to their healthy peers [41]. Moreover, the weight and mass distribution of exoskeletons directly affect energy expenditure [42,43,44]. Thus, it is crucial for exoskeletons designed for children with CP to incorporate a controller with a gravity compensator, decreasing energy expenditure and enhancing comfort. Moreover, dynamic modeling can be used to develop assist-as-needed control strategies, which provide the appropriate level of assistance based on the user’s abilities and needs. By continuously monitoring the user’s performance and adapting the level of assistance in real time, these control strategies can help reduce spasticity and improve motor function in children with CP.

This study considers the coupled human–exoskeleton system as an integrated system and tries to model the dynamics of the coupled human–SLE system [65,66,67]. The dynamic models of coupled human–LLEs are usually presented by the inverted pendulum approach, as shown in Figure 6 [68,69]. The robotic SLE can be considered a fixed-base mechanism; therefore, the dynamic modeling and control approaches of fixed-base manipulators can be efficiently employed [65]. The overground gait cycle is divided into two main phases: the swing phase and the stance phase. During the swing phase, the SLE can be simplified as a hanging triple pendulum, operating under the assumption that the hip remains stationary. This simplification allows for a more straightforward analysis and control design for the exoskeleton during this phase of the gait cycle. In the stance phase, on the other hand, the exoskeleton is modeled as an inverted pendulum, with the ankle joint acting as the fixation point. This modeling approach captures the inherent dynamics and stability characteristics of the exoskeleton during the weight-bearing portion of the gait cycle [54].

The Euler–Lagrange (E-L) and the Newton–Euler (N-E) formulations are commonly employed to model multibody dynamics. The former is based on energy, while the latter relies on forces to derive the dynamic model. The literature reveals that the E-L formulation has been extensively utilized to model the dynamics of exoskeletons [65]. The E-L equation is presented as follows [35]:(15)$$\frac{d}{dt}\frac{\partial L}{\partial \dot{q}}-\frac{\partial L}{\partial q}=\tau$$
the Lagrangian L is defined as L=K−P, where K represents the kinetic energy and P is the potential energy of the system. In this question, *q* denotes the variable of generalized coordinates, and τ represents the torque of the system. With (15), the dynamic model of a simplified coupled human–exoskeleton (Figure 6A) can be presented as follows [35,65]:(16)$$M\left(q\right)\ddot{q}+C\left(q,\dot{q}\right)\dot{q}+G\left(q\right)=\tau$$
where *M*(*q*) is the inertia matrix, including the exoskeleton inertia *M_e_*(*q*) and the human inertia *M_h_*(*q*). $C(q,\dot{q})$ represents the Coriolis and centripetal matrix, with $C_{e}(q,\dot{q})$ and $C_{h}(q,\dot{q})$ as the exoskeleton and human terms, respectively. *G*(*q*) is the gravity vector, encompassing *G_e_*(*q*) and *G_h_*(*q*) for the exoskeleton and the human gravity terms, respectively. Additionally, *q* is the generalized angular-joint-displacement vector, and *τ* is the torque vector, including *τ_e_* and *τ_h_* as the applied torque from the exoskeleton and the human, respectively. The inertia matrix (*M*(*q*)), the Coriolis and centripetal matrix ($C(q,\dot{q})$), and the gravity vector (*G*(*q*)) for a fixed-based robot with *n*-DOF can be achieved based on the following equations [35]:(17)$$M\left(q\right)=\sum_{i=1}^{n}\left(m_{i}J_{vi}^{T}J_{vi}+J_{wi}^{T}R_{0}^{i}{{I_{ci}R}_{0}^{i}}^{T}J_{wi}\right)$$
(18)$$C\left(q,\dot{q}\right)=\frac{1}{2}\sum_{i=1}^{n}\left(\frac{\partial M_{kj}}{\partial q_{i}}+\frac{\partial M_{ik}}{\partial q_{j}}-\frac{\partial M_{ij}}{\partial q_{k}}\right){\dot{q}}_{i}$$
(19)$$G\left(q\right)={\left(\begin{array}{ccc} \frac{\partial P}{\partial q_{1}} & \cdots & \frac{\partial P}{\partial q_{n}} \end{array}\right)}^{T}$$
where *m_i_* is the mass, *J_vi_* and *J_wi_* are linear and angular Jacobians, *R^i^* is the rotational matrix for the *i*-th link of the coupled human–SLE system, and *P* represents the potential energy. The extension of the equations for a 3-DOF coupled human–SLE system (Figure 6A) is provided in Appendix B. Equation (16) describes the forward dynamics of the coupled human–exoskeleton system. In contrast, the inverse dynamics of the system, which are particularly useful for simulation modeling, is presented below:(20)$$\ddot{q}={M(q)}^{-1}\left(\tau -C\left(q,\dot{q}\right)\dot{q}-G\left(q\right)\right)$$

The presented inverse dynamics model is directly implemented to model the coupled human–exoskeleton system in MATLAB Simulink^®^. This approach enables the simulation of the system’s behavior under different conditions and facilitates the analysis of its dynamics.

In this section, the dynamic model of the coupled human–exoskeleton system is derived using the transformation matrices discussed in the previous section. This helps enhance the understanding of the complex behavior of the coupled human–exoskeleton system and enables the design of a more effective control algorithm. As a result, smoother and more coordinated movements are achieved, preventing the triggering of spasms in CP users. This dynamic model effectively aids in designing the gravity compensator within the control system, considering the complex interaction between the user and the exoskeleton. The gravity compensator minimizes or eliminates the exoskeleton’s weight impact on the user, reducing energy expenditure and making the device feel lighter and more comfortable to wear. As mentioned earlier, children with CP have twice the metabolic consumption compared to their healthy peers [41]. Moreover, the weight and mass distribution of exoskeletons directly affect energy expenditure [42,43,44]. Thus, it is crucial for exoskeletons designed for children with CP to incorporate a controller with a gravity compensator, decreasing energy expenditure and enhancing comfort. In this scenario, the controlled SLE with gravity compensation functions as an assistive device with a zero-gravity effect.

A comprehensive dynamic model facilitates the development of exoskeletons’ adaptable to different users, walking patterns, and environments. This adaptability is essential for creating personalized devices catering to the individual needs of users with various physical conditions, particularly for exoskeletons designed for children with CP. Furthermore, this dynamic modeling approach enabled us to create simulations to analyze the exoskeleton’s behavior and refine its design before constructing a physical prototype. This effectively saved time and resources during the development process, helping identify potential issues before they become critical, ultimately increasing safety and reliability.

## 5. Controller Structure

As discussed in the previous section, the coupled human–SLE system can be simplified as a hanging triple pendulum during the swing phase and as an inverted pendulum during the stance phase. The inverted pendulum exhibits highly unstable dynamics, making it essential to implement a controller to ensure system stability and prevent the user and exoskeleton from falling.

The literature indicates that impedance control is a widely used control algorithm for LLEs, particularly for those designed for children with CP [11]. Impedance controllers are useful for controlling the interaction between the exoskeleton and the user’s limb, particularly in situations where the impedance of the limb is not well-defined, such as in individuals with neuromuscular disorders like CP. Impedance control enables exoskeletons to maintain compliant and adaptive interactions with users by regulating force and joint displacement, preventing injury from excessive force.

For children with CP, spasticity and varying GMFCS levels can present challenges in providing effective assistance. Impedance control is particularly effective in addressing these issues. By adjusting impedance parameters, the algorithm can be customized for different users and tasks, accommodating spasticity and varying GMFCS levels, ensuring safer and smoother interactions. Focusing on force control rather than solely on joint positions allows the exoskeleton to react to unexpected forces or changes in user movements, which is especially beneficial for users with spasticity. Impedance control also contributes to the stability of the human–SLE system and provides natural-feeling assistance by mimicking human muscle and joint characteristics. This natural interaction helps children with CP feel more comfortable and confident when using the exoskeleton, ultimately improving their overall experience and outcomes.

It is important to note that children with CP have metabolic consumption rates twice as high as those of their healthy peers [39,40,41], with energy consumption being directly related to the effects of gravity [42,43,44]. As such, the controller should not only ensure system stability but also compensate for gravity, making the controlled human–SLE system suitable for CP users who struggle with the burden of weight.

In general, the dynamic equation of a simplified coupled human–SLE with actuators can be written as follows:(21)$$M\left(q\right)\ddot{q}+C\left(q,\dot{q}\right)\dot{q}+B\dot{q}+G\left(q\right)=u$$
where *M* (*q*) represents the inertia matrix, which includes both the coupled system inertia and actuator inertia. Furthermore, *B* is negligible in comparison with $C(q,\dot{q})$, and for simplicity, we will take *B* = 0. When the input is not saturated, there is no significant difference in simulating with or without friction. Therefore, we obtain the following:(22)$$M\left(q\right)\ddot{q}+C\left(q,\dot{q}\right)\dot{q}+G\left(q\right)=u$$

According to the PD impedance controller with gravity compensation, we obtain the following control signal:(23)$$u=K_{p}\tilde{q}+K_{d}\dot{\tilde{q}}+G\left(q\right)$$
where $\tilde{q}$ is the angular position error between the joint references and the actual positions ($\tilde{q}=q_{ref}-q$), and $K_{p}$ and $K_{d}$ are positive diagonal matrices. Since $q_{ref}$ is assumed as a step input, $\dot{\tilde{q}}=-\dot{q}$ and ${\dot{q}}_{ref}=0$. According to (22) and (23), we obtain the following:(24)$$M\left(q\right)\ddot{q}+C\left(q,\dot{q}\right)\dot{q}+G\left(q\right)=K_{p}\tilde{q}-K_{d}\dot{q}+G\left(q\right)$$

After removing the gravity term ($G\left(q\right)$), we have:(25)$$M\left(q\right)\ddot{q}=K_{p}\tilde{q}-K_{d}\dot{q}-C\left(q,\dot{q}\right)\dot{q}$$

To prove that the designed controller in (23) is stable and results in asymptotic tracking, consider the following Lyapunov function candidate,
(26)$$V=\frac{1}{2}{\dot{q}}^{T}M\dot{q}+\frac{1}{2}{\tilde{q}}^{T}K_{p}\tilde{q}$$
*V* represents the total energy and is a positive function, except at the equilibrium position where $q=q_{ref}$ and $\dot{q}=0$; at this point, *V* is zero. If we can prove that *V* decreases during any motion, it will demonstrate that the system approaches the equilibrium position.
(27)$$\frac{d}{dt}\left(\frac{1}{2}{\tilde{q}}^{T}K_{p}\tilde{q}\right)=\frac{1}{2}2{\tilde{q}}^{T}K_{p}\frac{d}{dt}\left(q_{ref}-q\right)\;=-{\tilde{q}}^{T}K_{p}\dot{q}\overset{scaler\;value}{\to }={-\dot{q}}^{T}K_{p}\tilde{q}$$

From (26) and (27), we obtain the following:(28)$$\dot{V}={\dot{q}}^{T}M\ddot{q}+\frac{1}{2}{\dot{q}}^{T}\dot{M}\dot{q}-{\dot{q}}^{T}K_{p}\tilde{q}$$

From (28) and (25), we can obtain the following:(29)$$\dot{V}={\dot{q}}^{T}\left(K_{p}\tilde{q}-K_{d}\dot{q}-C\dot{q}\right)+\frac{1}{2}{\dot{q}}^{T}\dot{M}\dot{q}-{\dot{q}}^{T}K_{p}\tilde{q}\;=-{\dot{q}}^{T}K_{d}\dot{q}+\frac{1}{2}{\dot{q}}^{T}\left(\dot{M}-2C\right)\dot{q}$$

Since $\dot{M}-2C$ is skew-symmetric [35], ${\dot{q}}^{T}\left(\dot{M}-2C\right)\dot{q}=0$. Therefore, we obtain the following:(30)$$\dot{V}=-{\dot{q}}^{T}K_{d}\dot{q}\leq 0$$
where $K_{d}$ is a positive definite diagonal matrix. This result shows that *V* decreases as long as $\dot{q}$ is non-zero. Furthermore, it is important to demonstrate that once $\dot{V}$ becomes zero, we reach the equilibrium position ($q=q_{ref}$). To this end, suppose $\dot{V}=0$ for all times. Since $K_{d}$ is a positive definite matrix, $\dot{q}=0$ and hence $\ddot{q}=0$. By substituting this in (25), we find the following:(31)$$K_{p}\tilde{q}=0\overset{∆}{\to }\tilde{q}=0\overset{∆}{\to }q=q_{ref}$$

Finally, La Salle’s theorem [35] confirms that the model is a globally asymptotic stable system in the equilibrium position ($q=q_{ref}$). It should be noted that if the gravity term ($G\left(q\right)$) is unknown, this controller cannot be applied, and a robust controller must be designed to handle uncertainties.

## 6. Simulation and Validation Results

This section aims to simulate the coupled human–SLE system using the PD impedance controller and gravity compensator in MATLAB Simulink^®^ and Simscape^™^. Our objective is to validate the derived dynamics model of the coupled human–SLE system in the previous sections and demonstrate the controlled system stability. Figure 7 depicts the hierarchical control architecture for the coupled human–SLE system. Within this architecture, the supervisory-level controller provides the desired trajectory to the high-level controller. The high-level controller then employs the PD impedance controller with a gravity compensator to generate the required torque for the low-level controller. In turn, the low-level controller utilizes a PI controller to apply the appropriate control signal to the exoskeleton actuators.

We employed SolidWorks^®^ Student Edition (2021–2022) and Autodesk^®^ Fusion 360™ (2020) as the CAD software for the mechanical design and modeling of the SLE. Furthermore, we obtained the Q-6 dummy demo model from [70] and imported it into SolidWorks to represent the user. To model and analyze the dynamics of the coupled human–SLE system and establish a model-based controller, we exported the coupled human–SLE system (Figure 2) from SolidWorks^®^ and imported it into MATLAB^®^ Simscape. It should be noted that MATLAB^®^ R2021a was used in this study. A fixed-step size of $10^{-3}$ and an auto solver with a relative tolerance of $10^{-3}$ were employed to solve the differential equations.

The imported model in Simscape functioned as a black box system for us, meaning we had no knowledge of the system’s dynamic parameters. In order to design a model-based controller for the gravity compensator, we opted to model the system using Simulink. In this scenario, we successfully validated the derived dynamic model by comparing the Simulink results with the corresponding Simscape signals. However, due to the extreme complexity of the imported model in Simscape, we needed to simplify it prior to implementing it in Simulink. Therefore, a simplified leg, consisting of a thigh, shank, and foot connected by three revolute joints, was modeled in Simscape to represent the complex imported model. Then, we utilized this simplified system in Simscape to model the coupled human–SLE system in Simulink, employing the MATLAB Function. The controller was designed using a MATLAB Function in Simulink and subsequently applied to both Simscape and Simulink to stabilize the system, track the trajectory, and counteract the effects of gravity.

The general architecture of the interaction among SolidWorks, Simscape, and Simulink is depicted in Figure 8. This figure illustrates that the coupled human–SLE system is designed in SolidWorks and imported to Simscape. Furthermore, this figure shows that the Simscape model comprises two sub-models: the detailed model and the simplified model, with the latter being derived from the former. Subsequently, this simplified model is utilized for modeling the coupled human–SLE system in Simulink. Finally, the outcomes of the Simulink model and the detailed model in Simscape are compared to validate the derived dynamic model of the system.

In Simscape, we employed an “Inertia Sensor” to measure the inertia properties of each link, including mass ($m_{i}$), length ($l_{i}$), center of mass ($l_{ci}$), and moment of inertia in the z-axis ($I_{zzi}$), where the index *i* denotes the link number. The first link represents the thigh, the second link corresponds to the shank, and the third link demonstrates the foot, as depicted in Figure 6. The symbolic calculation of the E-L equation of motion in Appendix B reveals that only the moment of inertia in the *z*-axis of each link is included in the equations, while other inertia terms are eliminated. Consequently, we focused on measuring $I_{zzi}$ and disregarded the other inertia terms. The inertia properties for the human, SLE, and the detailed model of the coupled human–SLE system were measured in Simscape. These values were used in the simplified model within Simscape and the Simulink model to validate the derived dynamics model of the coupled human–SLE system and confirm its stability under the designed controller.

Table 4 presents the numerical values of the mass and the length of each link for both the human and the exoskeleton in Simscape. To enhance the accuracy and minimize the angular position error between the detailed model and the simplified model, we measured each parameter with 15 decimal places to ensure that the Simulink model is a reliable version of the detailed model. This study introduces a novel approach called the “Simscape Multibody Simplification Method” (SMS method), which provides a step-by-step guide for deriving a simplified model from a detailed model. The SMS method is detailed in Appendix D of this paper. According to the SMS method, in the inertia sensor, options for “Span Weld Joints”, “Mass”, “Center of Mass”, and “Centered Inertia Matrix” should be activated, with the “Measurement Frame” set to “Custom”. The configuration of the inertia sensor for the hip connection between the detailed model and the simplified model within Simscape is illustrated in Figure 9. Table 5 presents the inertia properties of the detailed model and the simplified model of the coupled human–SLE system in Simscape, enabling comparison. In Table 5, the measured mass of each link ($m_{i}$) for the detailed model is equal to the sum of the user’s mass and the SLE’s mass, as presented in Table 4. Moreover, the measured mass for each link in the simplified model is exactly half that of the detailed model. The same scenario applies to the moment of inertia in the z-axis ($I_{zzi}$), where the measured inertia of the simplified model is exactly half that of the detailed model. The center of mass ($l_{ci}$) is exactly the same for both the detailed model and the simplified model, where the length of each link ($a_{i}$) is approximately equal between the two models.

According to Table 5, the numerical values of the inertia matrix (*M*(*q*)), Coriolis and centripetal matrix ($C(q,\dot{q})$), gravity vector (G(q)), and torque vector (τ) for a 3-DOF coupled human–SLE system (Figure 6A) are provided in Appendix B. The Detailed-Model and the Simplified-Model of the coupled human–SLE system, along with their configurations within Simscape, are depicted in Figure 10. Figure 10A displays the main elements of the detailed model, imported from SolidWorks, while Figure 10B presents the components used in the simplified model. The inputs for the models are torque, and the outputs consist of joint angles, including those of the hip, knee, and ankle. According to Table 5, the mass and inertia of the detailed model are twice those of the simplified model. As a result, it is expected that the torque applied to the detailed model is double the torque applied to the simplified model. This is the reason why the torques applied to the detailed model are multiplied by gains with values of two. 

Figure 11 depicts the controlled human–SLE system in Simulink, where the green box represents a MATLAB function encompassing the inverse dynamics of the coupled human–SLE system. The green box has inputs for torque, angular position, and angular velocity and outputs the angular acceleration for each joint of the hip, knee, and ankle. The green box contains the numerical values of parameters for the inertia matrix (*M*(*q*)), the Coriolis and centripetal matrix ($C(q,\dot{q})$), and the gravity vector (*G*(*q*)) as described in (20) for the inverse dynamics of the 3-DOF coupled human–SLE system, provided in Appendix C. As depicted in Figure 11, the gray box labeled “Desired Leg Angles” generates the reference angular positions, which are subsequently fed into the yellow box labeled “Controller”. In addition to the reference trajectories, the yellow box (controller) receives angular positions and angular velocities as inputs, and it utilizes the control laws in (23) to generate desired torque commands, driving the coupled human–SLE system toward the desired angular positions. The controller architecture, given by the equation $u=K_{p}\tilde{q}+K_{d}\dot{\tilde{q}}+G\left(q\right)$ in (23), is displayed in Figure 12. As discussed in Section 5, $K_{p}$ and $K_{d}$ are positive diagonal matrices representing the proportional and derivative gains, respectively, which contribute to the control input. The position error, denoted by $\tilde{q}$, and its rate of change, denoted by $\dot{\tilde{q}}$, are multiplied by these gain matrices. The term $G\left(q\right)$ represents gravity compensation. In Figure 12, the proportional gain matrix $K_{p}$ has values of 80, 80, and 50 along its main diagonal, while the derivative gain matrix $K_{d}$ has values of 12, 12, and 2 along its main diagonal.

The green box utilizes the torque provided by the controller and the angular position and angular velocity as feedback to compute the angular acceleration for each joint, employing the inverse dynamics technique in (20) for the coupled human–SLE system. By using this approach, the green box can accurately simulate the behavior of the system and provide insights into its dynamics. After integrating the angular acceleration computed by the green box a couple of times, the angular velocity and angular position for each joint are obtained. Since the computed angular positions are in radians, we used gains with a rate of 180/π to convert them from radians to degrees, as shown in Figure 13.

The angular positions of the reference (*θ*_Ref_), the detailed model in Simscape (*θ*_Simscape_), and the Simulink model (*θ*_Simulink_) for each joint of the hip, knee, and ankle for two gait cycles are shown in Figure 13A–C. The reference value changes every three seconds, with each gait cycle being completed within six seconds. In these figures, the Simulink and Simscape angular positions precisely overlap for all hip, knee, and ankle joints. Figure 13D displays the angular position error between the detailed model (Figure 10A) and the simplified model (Figure 10B) in Simscape, indicating that the error range is within $10^{-13}$ degrees for all hip, knee, and ankle joints. This error range demonstrates the reliability of the simplified model in Simscape, which will be used to design the Simulink model for the coupled human–SLE system. Figure 13E presents the discrepancy in angular positions between the detailed model in Simscape (Figure 10A) and the Simulink model (Figure 11), with the error margin being within $10^{-11}$ degrees for all hip, knee, and ankle joints. This level of error validates the accuracy of the Simulink model, which will be employed for the development of the gravity compensator (Figure 12) in the control system. Furthermore, Figure 13E serves to validate the accuracy of the information provided in Appendix B and Appendix C.

The discrepancy between the reference angular positions (generated by the gray box in Figure 11) and the controlled detailed model in Simscape (depicted in Figure 10A) is illustrated in Figure 13F. The error range reaches $10^{-6}$ after three seconds for the angular positions of all joints, confirming the accuracy of the derived dynamic model and the controller’s effectiveness. In Figure 11, the final output from the green box, labeled “Verification”, serves to confirm the accuracy of the derived dynamic model for the coupled human–SLE system, presented in Figure 14. Figure 14 illustrates the numerical values of ($q^{T}\left(\dot{D}\left(q\right)-2C(q,\dot{q})\right)q$) over a couple of gait cycles, falling within the range of $10^{-16}$ and being approximately zero. If the dynamic model is derived accurately, the numerical values of ($q^{T}\left(\dot{D}\left(q\right)-2C(q,\dot{q})\right)q$) are expected to be zero, in accordance with the properties of skew-symmetric matrices $\dot{D}\left(q\right)-2C(q,\dot{q})$ [35]. This figure serves as confirmation of the high accuracy achieved in deriving the dynamic model for the coupled human–SLE system.

Figure 15 depicts the control signals for the hip, knee, and ankle joints of the coupled human–SLE system. The key observation from this figure is that due to the presence of the gravity vector (*G*(*q*)), the control signal during the settling time is non-zero. This non-zero control signal is essential for counteracting the gravitational effects on each link of the system. Furthermore, it is important to note that these control signals are considered safe for use with the actuators in this assistive device.

Figure 16 presents a three-dimensional isometric visualization of the human–SLE system during a gait cycle, spanning from 6 s to 12 s. This gait cycle is divided into two key phases: the stance phase and the swing phase. The stance phase, occurring between 6 s and 9 s, demonstrates the period when the foot is in contact with the ground, providing support and stability. Conversely, the swing phase, which takes place from 9 s to 12 s, captures the time when the foot is in motion, swinging forward to prepare for the next step. This comprehensive representation offers valuable insights into the biomechanics of human gait and the role of the SLE system in facilitating effective locomotion. Additionally, a Appendix A is provided alongside this paper, visually demonstrating the simulation of the human–SLE system while walking through two gait cycles over a duration of 12 s.

## 7. Conclusions

This study designed and controlled a novel, size-adjustable exoskeleton tailored for children aged 8–12 with cerebral palsy (CP) across GMFCS levels I–IV. It analyzed a range of body measurements to accommodate the varied physical dimensions of this group. Brushless DC motors and harmonic drive gears were chosen for their efficiency, and an EtherCAT communication protocol was employed to enable high-frequency, responsive operation. The exoskeleton, managed by a high-performance embedded controller, is equipped with distinctive mechanical, sensory, and control features, offering personalized support considering the wide range of muscle tone and spasticity in children with CP. This study used the Denavit–Hartenberg matrix method and a geometric approach to examine the exoskeleton’s kinematics. A dynamic model of the coupled human–exoskeleton system was created to design a PD impedance controller with gravity compensation. The system was designed in SolidWorks, then imported into Simscape in MATLAB, known as the “Detailed-Model”. A “Simplified-Model” was also developed to provide similar trajectories, using a novel methodology introduced in this study, namely the “Simscape Multibody Simplification Method”. The study modelled an independent coupled human–exoskeleton system in Simulink using the Euler–Lagrange formulation and a PD impedance controller with a gravity compensator, which was then applied to three models within the same environment. The minimal discrepancies between the models validated the dynamic model and confirmed the controller’s effectiveness. The derived results show that the dynamic model was verified to follow the predefined gait patterns with high accuracy using a hardware-in-the-loop setup.

## Figures and Tables

**Figure 1 sensors-23-06103-f001:**
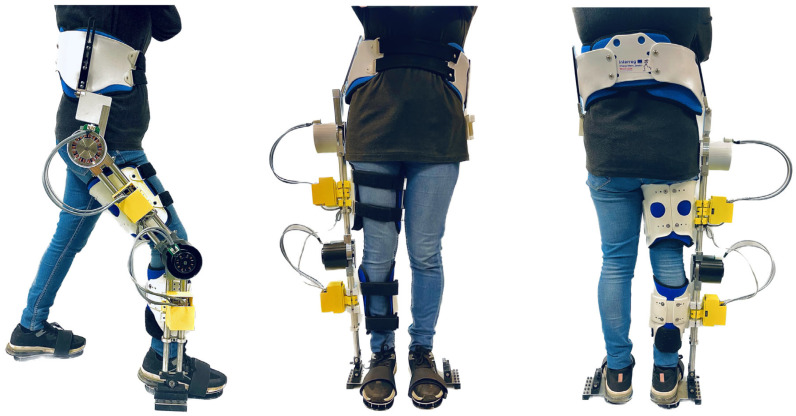
The prototyped SLE with two actuated joints at the hip and knee from side, front, and back perspectives.

**Figure 2 sensors-23-06103-f002:**
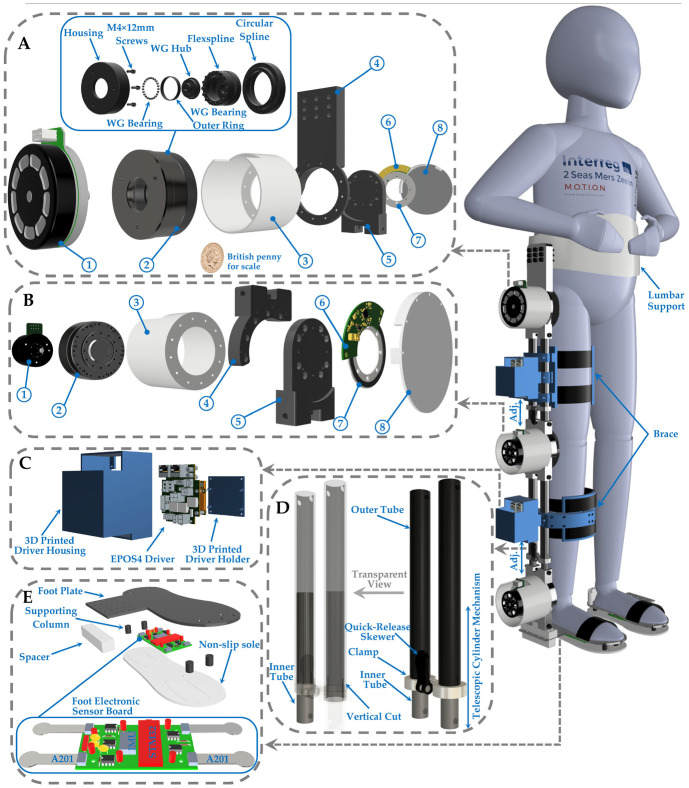
This is an overview of a novel single-leg exoskeleton designed to improve walking functionality in children with CP. (**A**) Exploded view of hip actuator assembly. (1) A 220 W EC 90 flat Maxon Motor. (2) A strain-wave gearbox with 100:1 reduction ratio (CPU-17A-100-M-10.15). (3) 3D-printed motor housing. (4) Hip actuator upper joint. (5) Hip actuator lower joint. (6) Rotary absolute magnetic encoder readhead. (7) Encoder magnetic ring. (8) 3D-printed encoder holder. (**B**) Knee actuator assembly overview. (1) A 150 W EC 60 flat Maxon Motor. (2) A gearbox with 50:1 reduction ratio (CPU-17A-50-M-8.49). (3) 3D-printed motor housing. (4) Hip actuator upper joint. (5) Hip actuator lower joint. (6) Rotary absolute magnetic encoder readhead. (7) Encoder magnetic ring. (8) 3D-printed encoder holder. (**C**) EPOS4 Driver structure. (**D**) Lower-leg tubes and transparent view. (**E**) Foot configuration.

**Figure 3 sensors-23-06103-f003:**
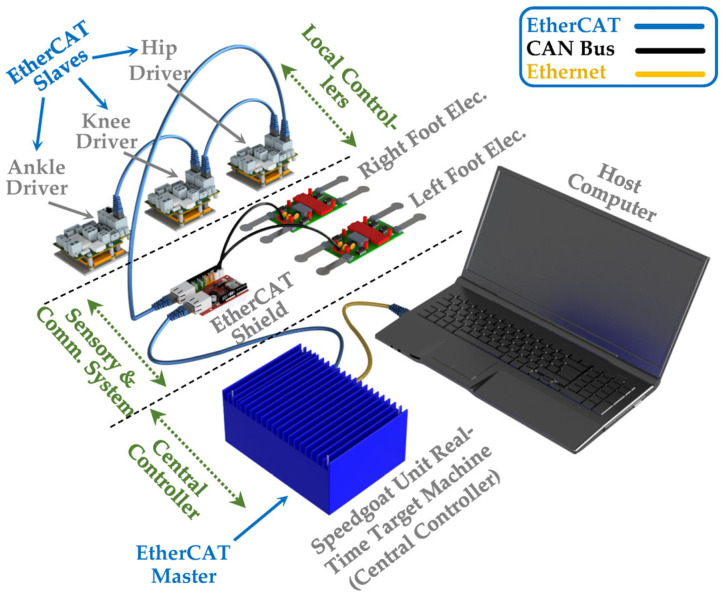
Control system configuration and communication architecture of the SLE.

**Figure 4 sensors-23-06103-f004:**
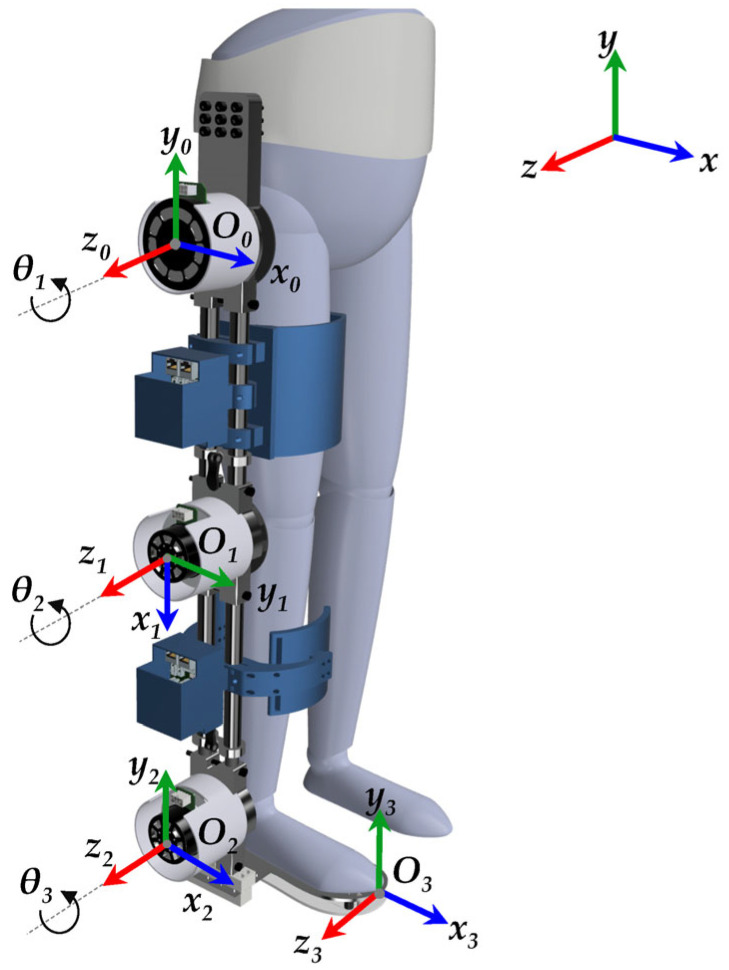
D-H Coordinate frames attached to the SLE.

**Figure 5 sensors-23-06103-f005:**
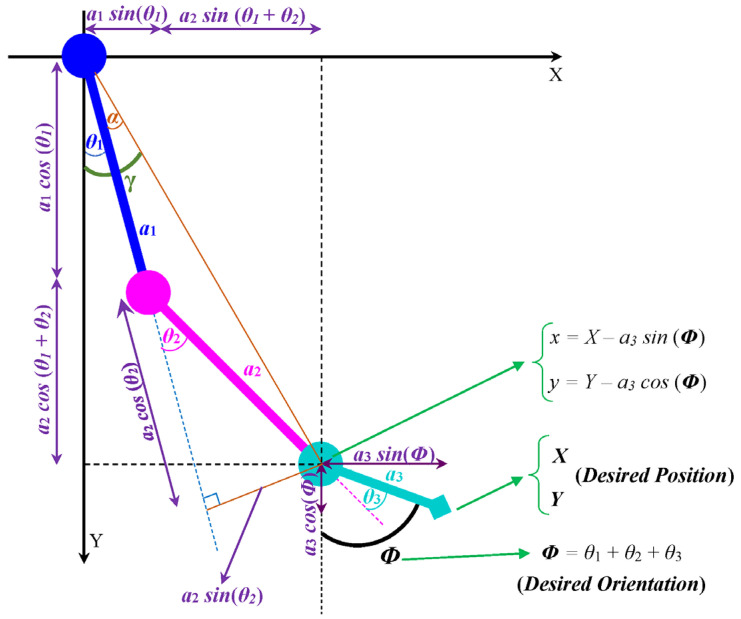
Projection of the SLE onto X–Y plane.

**Figure 6 sensors-23-06103-f006:**
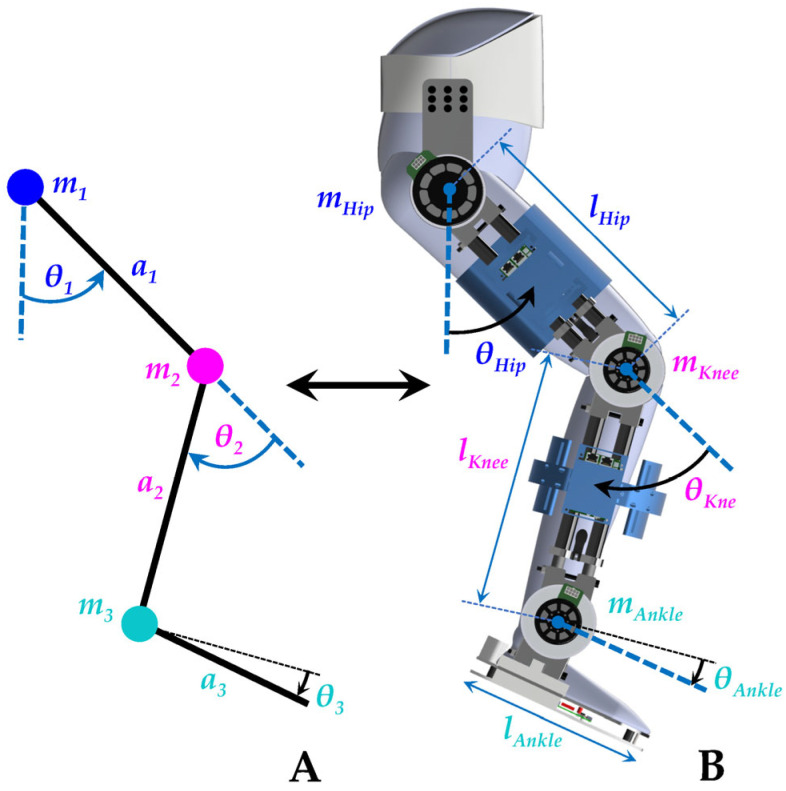
The dynamic model of the coupled human–SLE is based on the inverted pendulum. (**A**) Simplified model. (**B**) Coupled human–SLE’s dimensions and mass distribution.

**Figure 7 sensors-23-06103-f007:**
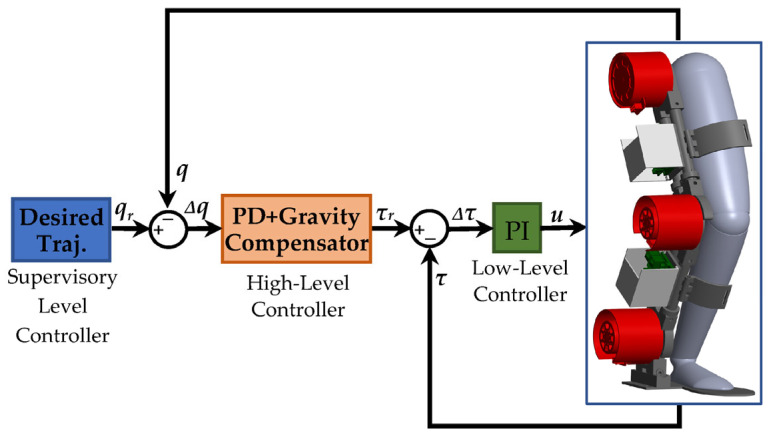
Controller architecture for the coupled human–SLE system.

**Figure 8 sensors-23-06103-f008:**
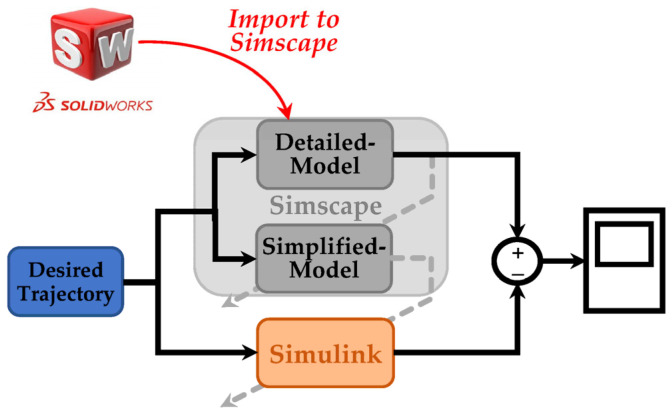
General scheme in simulation section.

**Figure 9 sensors-23-06103-f009:**
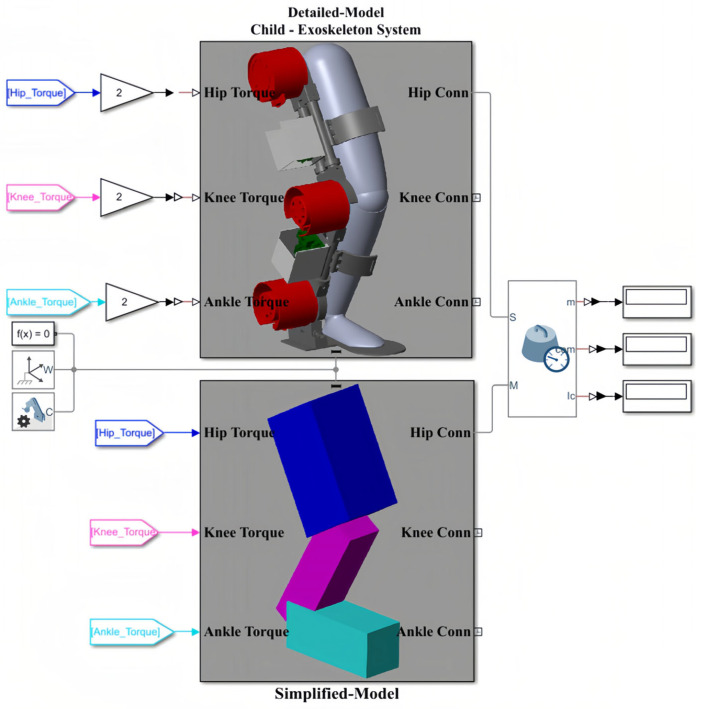
Inertia Sensor connection to the detailed model and the simplified model of the coupled human–SLE system in Simscape.

**Figure 10 sensors-23-06103-f010:**
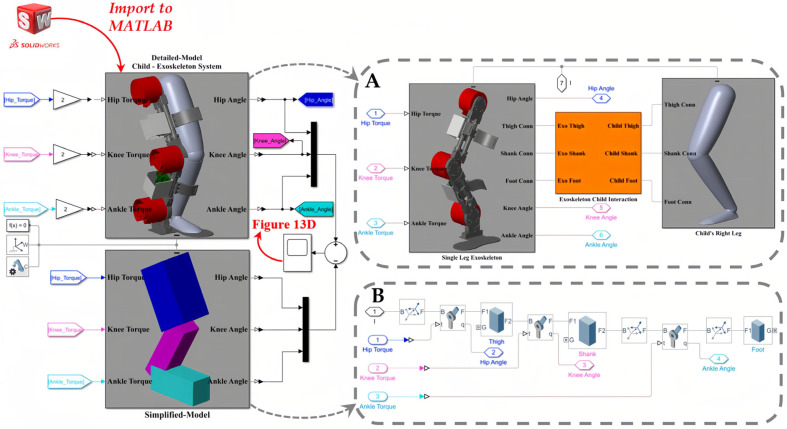
The general model of the coupled human–SLE system in Simscape. (**A**) The detailed model, imported from SolidWorks into Simscape, includes the SLE, human leg, and connection section. (**B**) The simplified model, consisting of elements in Simscape.

**Figure 11 sensors-23-06103-f011:**
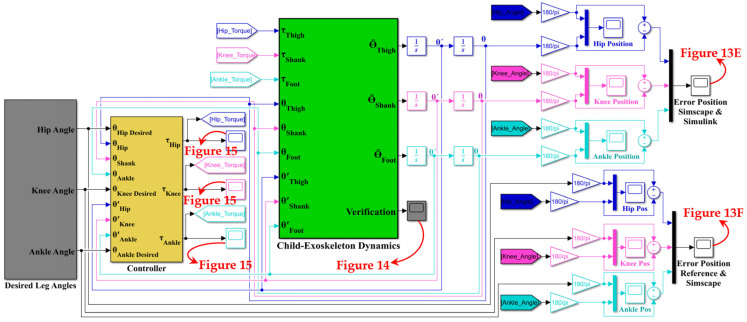
Model of the coupled human–SLE system in Simulink.

**Figure 12 sensors-23-06103-f012:**
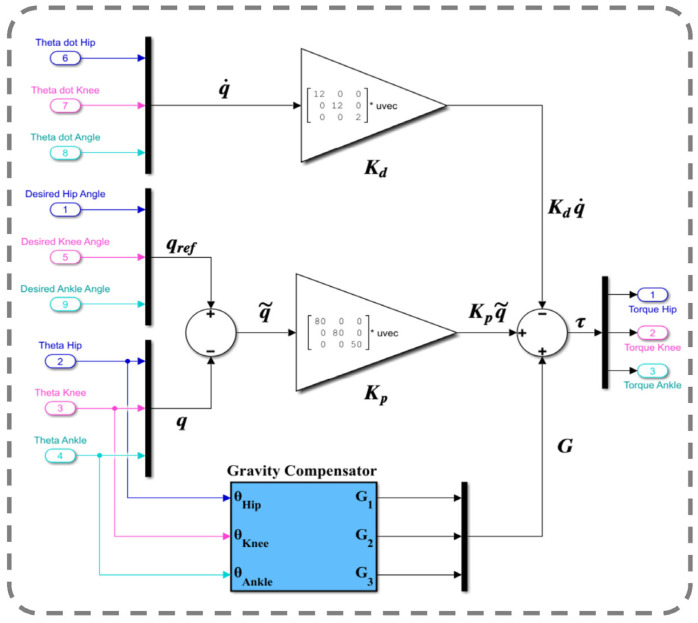
PD impedance controller with a gravity compensator.

**Figure 13 sensors-23-06103-f013:**
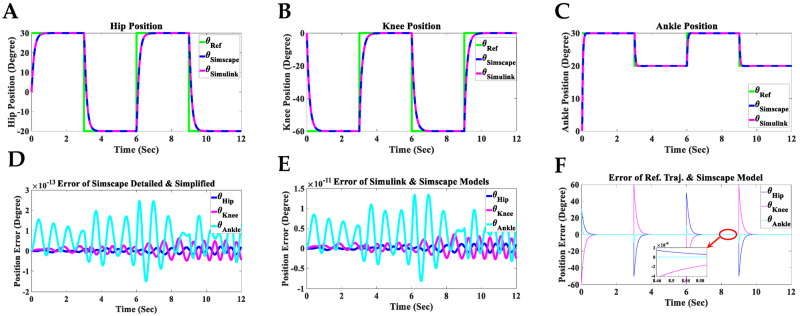
Simulation results of the controlled coupled human–SLE system in Simscape and Simulink. (**A**) Hip position, (**B**) knee position, (**C**) ankle position, (**D**) position error between the Simscape detailed model (imported model) and the Simscape simplified model, (**E**) position error between the Simulink model and the Simscape detailed model, (**F**) position error between the reference trajectory and the Simscape detailed model.

**Figure 14 sensors-23-06103-f014:**
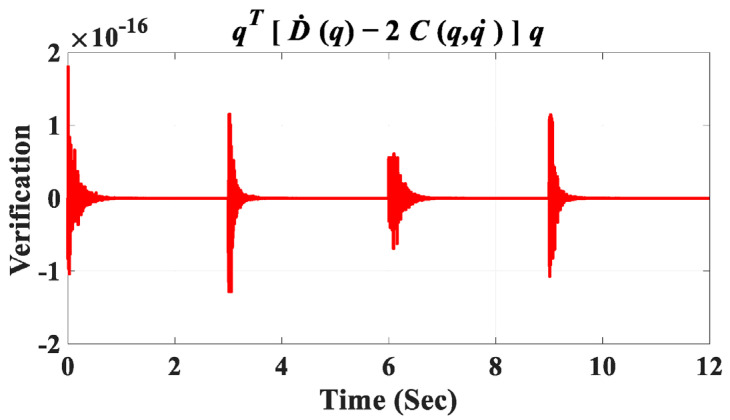
Verification signal.

**Figure 15 sensors-23-06103-f015:**
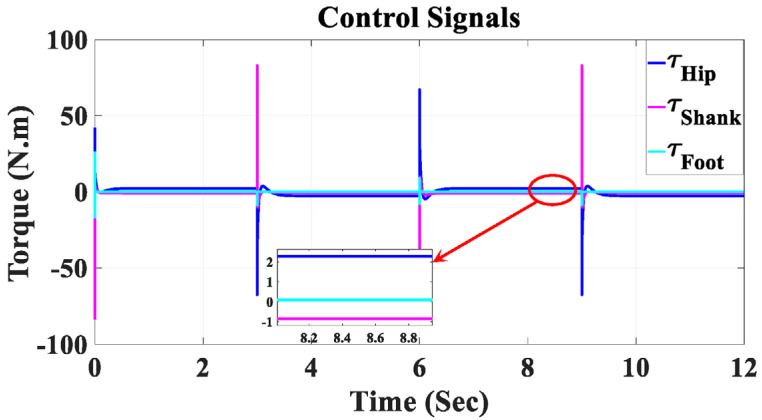
Control signals for the hip, shank, and foot joints in Simulink.

**Figure 16 sensors-23-06103-f016:**
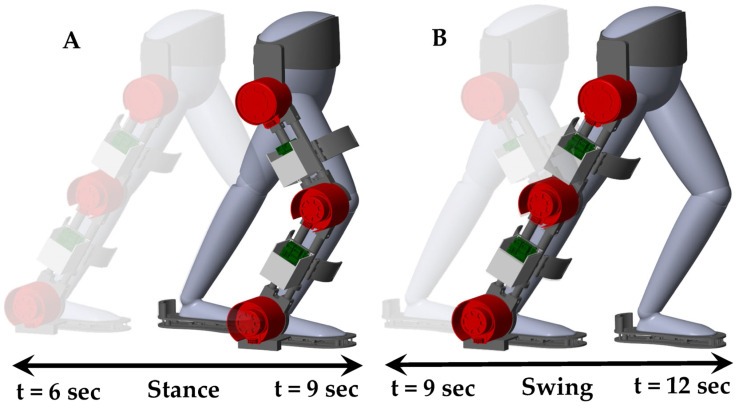
Three-dimensional isometric representation of the human–SLE system throughout a gait cycle from 6 s to 12 s, subdivided into stance and swing phases. (**A**) Depicts the stance phase occurring between 6 s and 9 s, and (**B**) illustrates the swing phase from 9 s to 12 s.

**Table 1 sensors-23-06103-t001:** Average body dimensions of CP and TD children, along with the minimum and maximum of the SLE dimensions.

Type	CP Children [45]		TD Children [49]		Exoskeleton	
Gender	Girls	Boys	Girls	Boys	*-*	*-*
Percentile	50th	50th	50th	50th	*-*	*-*
Age (years)	8	12	6	12	*-*	*-*
GMFCS (level)	IV	I	*-*	*-*	min	MAX
H (cm)	110 [47]	143 [48]	112	141	78	104
LL (cm)	62.8 ^1^	85.7 ^2^	63.9	84.5	60	86
FL (cm)	16.5 ^1^	21.7 ^2^	16.8	21.4	23.5	23.5
ULL (cm)	-	-	-	-	27	40
LLL (cm)	-	-	-	-	33	46

H: height; LL: leg length; FL: foot length; ULL: upper-leg length; LLL: lower-leg length. ^1^ Considering the height of a girl with CP and GMFCS level IV to be 110 cm [47], which is similar to the height of a healthy 6-year-old girl (112 cm) [49] (p. 524), we can estimate the leg length (LL) and foot length (FL) using the proportion of LL and FL from a healthy 5-year-old girl: LL = (110 × 63.9)/112 = 62.8 cm and FL = (110 × 16.8)/112 = 16.5 cm. ^2^ Considering the height of a boy with CP and GMFCS level I to be 143 cm [48], which is similar to the height of a healthy 12-year-old boy (141 cm) [49] (p. 524), we can estimate the leg length (LL) and foot length (FL) using the proportion of LL and FL from a healthy 12-year-old boy: LL = (143 × 84.5)/141 = 85.7 cm and FL = (143 × 21.4)/141 = 21.7 cm.

**Table 2 sensors-23-06103-t002:** Actuation unit configuration for each joint.

Actuated Joint	Weight (kg)	Power Rate (W)	Reduction Ratio	Max. Torque (Nm)	Max. Velocity (deg/s)
Hip	0.624	220	100:1	150	300
Knee	0.35	150	50:1	70	720
Ankle	0.35	150	120:1	168	300

**Table 3 sensors-23-06103-t003:** D-H robotic SLE parameters.

Link	*θ_i_*	*d_i_*	*a_i_*	*α_i_*
1	*θ_1_* − *π*/2	0	*a* _1_	0
2	*θ_2_*	0	*a* _2_	0
3	*θ_3_* + *π*/2	0	*a* _3_	0

**Table 4 sensors-23-06103-t004:** Inertia properties of the human and SLE in Simscape.

Parameter	Unit	Human (User)	SLE
$m_{1}$	kg	1.990395854171116	2.534550867105971
$m_{2}$	kg	0.846719812062487	2.216669253523135
$m_{3}$	kg	0.220456270631801	1.973399444768174
$a_{1}$	m	0.304170	0.301100
$a_{2}$	m	0.319502	0.320201
$a_{3}$	m	0.217621	0.235000

**Table 5 sensors-23-06103-t005:** Inertia properties of detailed and simplified models of the coupled human–SLE system in Simscape.

Parameter	Unit	Detailed Model	Simplified Model
$m_{1}$	kg	4.524946721277088	2.262473360638544
$m_{2}$	kg	3.063389065585623	1.531694532792812
$m_{3}$	kg	2.193855715399975	1.096927857699987
$a_{1}$	m	0.301100	0.301100
$a_{2}$	m	0.320201	0.320201
$a_{3}$	m	0.23500	0.235600
$I_{zz1}$	kg·m^2^	0.036044993739971	0.018022496869986
$I_{zz2}$	kg·m^2^	0.023842199764414	0.011921099882207
$I_{zz3}$	kg·m^2^	0.001639377961393	0.000819688980696
$l_{cx1}$	m	−0.046206480105565	−0.046206480105565
$l_{cy2}$	m	0.000157578476401	0.000157578476401
$l_{cz3}$	m	0.046955758713415	0.046955758713415
$l_{cx2}$	m	−0.071976964086664	−0.071976964086664
$l_{cy2}$	m	−0.000300891583615	−0.000300891583615
$l_{cz2}$	m	0.062795593711711	0.062795593711711
$l_{cx3}$	m	−0.086892122966857	−0.086892122966857
$l_{cy3}$	m	−0.029380769989573	−0.029380769989573
$l_{cz3}$	m	0.009494757199261	0.009494757199261

## Data Availability

Data generated during the course of this study will be made available in subsequent publications.

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
