# Peer review of "Design and Control of a Single-Leg Exoskeleton with Gravity Compensation for Children with Unilateral Cerebral Palsy"

_sensors, 2023, doi:10.3390/s23136103_

Round 1

Reviewer 1 Report

In this paper, the authors propose the design and control system of a single-leg exoskeleton with gavity compensation for children with cerebral palsy.

The article has some interesting aspects but some revisions are needed to improve the content and legibility of the manuscript.

In the introduction, the authors should focus more on the literature gap they intend to fill by listing all pediatric exoskeletons for CP.

In this context, the following articles dealing with the design and testing of pediatric exoskeletons for CP are suggested.

Mileti, I., Taborri, J., Rossi, S., Petrarca, M., Patanè, F., & Cappa, P. (2016, May). Evaluation of the effects on stride-to-stride variability and gait asymmetry in children with Cerebral Palsy wearing the WAKE-up ankle module. In 2016 IEEE International symposium on medical measurements and applications (MeMeA) (pp. 1-6). IEEE.

Patane, F., Rossi, S., Del Sette, F., Taborri, J., & Cappa, P. (2017). WAKE-Up exoskeleton to assist children with cerebral palsy: design and preliminary evaluation in level walking. IEEE Transactions on Neural Systems and Rehabilitation Engineering, 25(7), 906-916.

Please add a reference to these sentences:

The promising clin- 83 ical results of this assistive device demonstrate that the use of a unilateral exoskeleton 84 seems to have the potential to be an effective robotic solution for individuals with one- 85 sided mobility impairments, such as children with hemiplegia CP

Model-based control is widely used in many applications but it requires a precise dynamic 96 model of the coupled human-exoskeleton system.

Please justify these sentences:

Relying on human-exoskeleton dynamic models, model-based control strategies can 106 be used for gravity compensation and balance control.

Please report the total weight of the exoskeleton: The exoskeleton weight and mass distribution directly af- 117 fect metabolic consumption and functional performance

The height and weight of the subject in the picture should be reported: In Figure 1, the SLE is worn by a 164 healthy adult with body dimensions similar to those of children.

The following sentences are just speculation. In this study author did not verify that the proposed exoskeleton effectively reduced the fatigue and the metabolic consumption on a population with CP. authors should remove all speculative phrases that are not actually verified by literature findings or actual results on populations with CP

“The author should  The SLE is made of lightweight and durable materials like Aluminum tubes to reduce 215 the overall weight of the device, reducing fatigue and metabolic consumption, which are 216 critical for children with CP, and making it easier for them to move around and maintain 217 their balance for extended periods”

A comparison table comparing all the mechanical characteristics of the proposed exoskeleton and actuation system with other exoskeletons in the literature and commercial exoskeletons should be introduced in the mechanical design section

it is unclear how asymmetry can be measured when the sensor system is present on only one limb: “This information can be utilized to evaluate gait patterns and 388 detect any asymmetries”

the sensor system should be specified by reporting the technical characteristics of all elements.

Please add a reference to this sentence: Tethered 463 power assistance eliminates the need for heavy batteries, leading to lighter devices that 464 are easier for children with CP to manage and maneuver

Even this is speculation unless supported by verification on population with CP : “By embracing tethered power assistance in SLE 473 devices, clinicians and therapists can more effectively address the specific needs of this 474 population, facilitating improved motor function and promoting a greater sense of auton- 475 omy.”

For branch schematics should be reported for the DH model detailing Branch, Parent, Children, Transforms. It is not clear whether floating base system or a fixed base system.

Please specify the ai in the DH table

The article has many redundant sentences that do not add quantitative information to the text and worsen the readability of the article. Authors should summarize the discussion of the materials and methods part by reporting only that quantitative and technical information necessary for the reproducibility of the content. In addition, there are many sentences that are repeated several times:  “In the stance phase, the exoskeleton is considered an inverted 727 pendulum, where the ankle joint serves as the fixation point”

The authors should justify this choice. : “To enhance the accuracy and minimize the error between the Detailed-Model and Simplified-Model, we measured each parameter with 15 decimal places”

What is the accuracy of the measurement system that allows accurate measurement beyond micrograms and beyond micrometers?

To improve the readability of the article, the authors would like to divide this paragraph into two parts, one in which they summarize the simulation carried out and the other the results.

There is currently a lack of a discussion paragraph in which they compare their results with literature results. The conclusion part should report the most significant results with the authors' considerations rather than a summary of the work. In this form, the article while interesting does not have sufficient readability and the message is not effectively reported.

Author Response

Please find attached the response letter we have crafted for the first reviewer's comments and suggestions.

Reviewer 2 Report

A meaningful work for disabled children (CP), and a systemic work was proposed. Thanks the authors for the efforts

1、 The paper said the efficiency was heightened by the use of brushless motor and harmonic drive gears. Is there any data to show this point? As I know, the efficiency of Harmonic Drive gears is not very high.

2、 What accuracy of the inverted pendulum?

3、 The power consumption of CP children is two times higher than heathy children. This statement was mentioned several times. However, there were no actual statistic data in the paper. We like to the actual test.

4、 Reference 1-5 were too old. Please add some new ones.

5、 As mentioned in 3, we strong suggest actual and practical test and data on the Exoskeleton.

6、 The paper was too long, and it is hard to read such a long work. Please reduce unnecessary content.

Author Response

Please find attached the response letter we have crafted for the second reviewer's comments and suggestions.

Reviewer 3 Report

Mechanical design as well as the design of the control (hardware and software) of a special single-leg exoskeleton are presented in this paper. Exoskeleton is intended to be used by children with the unilateral cerebral palsy. Since such patients consume more metabolic energy during walking it is important to reduce gravitational loads so novel control strategy which includes gravity compensation is proposed,

Paper is well structured and organized. Presentation is comprehensible and easy to follow. Experiment is explained in detail and results are clearly presented.

In order to improve the paper, I suggest to the authors to add some comments about further research, development of the prototype etc.

Author Response

Please find attached the response letter we have crafted for the third reviewer's comments and suggestions.

Round 2

Reviewer 1 Report

The authors responded to all requests by incorporating revisions into the manuscript.